# Revisiting Multi-Task Learning with ROCK: a Deep Residual Auxiliary Block for Visual Detection

**Taylor Mordan**[(1, 2)]
taylor.mordan@lip6.fr

**Nicolas Thome**[(3)]
nicolas.thome@cnam.fr

**Gilles Henaff**[(2)]
gilles.henaff@fr.thalesgroup.com

**Matthieu Cord**[(1)]
matthieu.cord@lip6.fr

(1) Sorbonne Université, CNRS, Laboratoire d'Informatique de Paris 6, LIP6
F-75005 Paris, France

(2) Thales Land and Air Systems
2 Avenue Gay-Lussac, 78990 Élancourt, France

(3) CEDRIC, Conservatoire National des Arts et Métiers
292 Rue St Martin, 75003 Paris, France

## Abstract

Multi-Task Learning (MTL) is appealing for deep learning regularization. In this paper, we tackle a specific MTL context denoted as *primary MTL*, where the ultimate goal is to improve the performance of a given primary task by leveraging several other auxiliary tasks. Our main methodological contribution is to introduce ROCK, a new generic multi-modal fusion block for deep learning tailored to the primary MTL context. ROCK architecture is based on a residual connection, which makes forward prediction explicitly impacted by the intermediate auxiliary representations. The auxiliary predictor's architecture is also specifically designed to our primary MTL context, by incorporating intensive pooling operators for maximizing complementarity of intermediate representations. Extensive experiments on NYUv2 dataset (object detection with scene classification, depth prediction, and surface normal estimation as auxiliary tasks) validate the relevance of the approach and its superiority to flat MTL approaches. Our method outperforms state-of-the-art object detection models on NYUv2 dataset by a large margin, and is also able to handle large-scale heterogeneous inputs (real and synthetic images) with missing annotation modalities.

## 1   Introduction

The outstanding success of ConvNets for image classification in the ILSVRC challenge [26] has heralded a new era for deep learning. A key element of this success is the availability of large-scale annotated datasets such as ImageNet [40]. When dealing with smaller-scale datasets, however, training such big ConvNets is not viable, due to strong overfitting issues. In some applications where images themselves are difficult to obtain, *e.g.* medical or military domains, getting additional annotations on available images can be easier than collecting more examples, as a way to get more data to feed the networks. An appealing option to limit overfitting is then to rely on Transfer Learning (TL), which aims at leveraging different objectives and datasets for improving predictive performances. The most popular strategy for tackling small datasets in vision is certainly Fine-Tuning (FT) [1], which

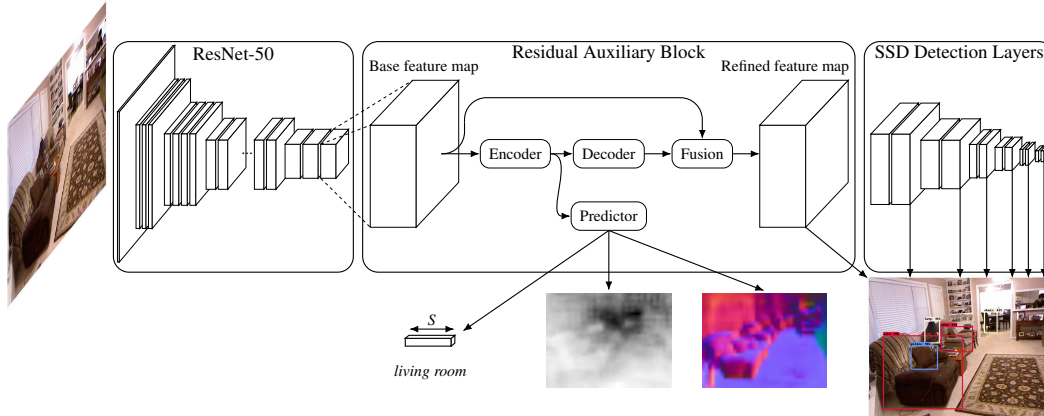

Figure 1: **Residual auxiliary block (ROCK) for object detection with auxiliary information.**
ROCK (middle) is incorporated into a backbone SSD object detection model [31] to utilize additional
supervision from multi-modal auxiliary tasks (scene classification, depth prediction and surface
normal estimation) and to improve performance on the primary object detection task.

can be regarded as sequential TL. It consists in using networks pre-trained on a large-scale dataset (*e.g.*
ImageNet), which provide very powerful visual descriptors known as Deep Features (DF). DF are at
the core of state-of-the-art dense prediction methods, *e.g.* object detection [17, 11, 16, 12, 31, 7, 36],
image segmentation [10, 5], depth prediction [28, 29, 14] or pose estimation [33].

A drawback with FT is that the large-scale data and labels are only used to initialize network
parameters, but not for optimizing the ultimate model used for prediction. At the other extreme,
Multi-Task Learning (MTL) solutions consist in using different tasks and datasets and to share some
intermediate representations between the tasks, which are optimized jointly [35, 3, 32, 25, 50, 34],
and can be seen as parallel TL. Although MTL is an old topic in machine learning [4], this is currently
intensively revisited with deep learning. The crux of the matter is to define where and how to share
parameters between tasks, depending on the applications. Some approaches focus on learning the
optimal MTL architectures [35, 32], while other explore relating every layer of the networks [50]
or to relate layers at various depths to account for semantic variations between modalities [34]. In
UberNet [25], the goal is to learn a universal network which can share various low- and high-level
dense prediction tasks [25]. This MTL strategy has been shown to improve results of individual tasks
when learned together under certain conditions.

The aforementioned approaches assume a flat structure between tasks, the goal of MTL usually
being to have good results on all tasks simultaneously while saving computations or time. However,
our problem is concerned with a primary task, which is augmented during training with several
auxiliary tasks. The ultimate goal is here to improve the primary task performance, not to have good
performances on average across tasks. Flat MTL is therefore intrinsically sub-optimal here, since the
problem is biased toward a given application. We frame it as a new kind of MTL, named primary
MTL, where there is only one task of interest, and other auxiliary tasks that can be leveraged to
improve the first, primary one. In this sense, our context is related to Learning Using Privileged
Information (LUPI) [47, 38, 41, 42, 46, 23, 43] and end-to-end trainable deep LUPI approaches.

In this paper, we introduce a new model for leveraging auxiliary supervision and improving perfor-
mance on a primary task, in a primary MTL setup. Regarding methodology, the main contribution
of the paper is to introduce ROCK, a new residual auxiliary block (Figure 1), which can easily be
inserted into any existing architecture to effectively exploit additional auxiliary annotations. The
main goal is to produce predictions for auxiliary tasks and to learn features through MTL. However, it
is designed around two key features differentiating it from flat MTL in order to better fit ROCK to our
context. First, the block is equipped with a residual connection, which explicitly merges intermediate
representations of the primary and auxiliary tasks, making the latter ones have a real effect on the
former in the forward pass, not just through shared feature learning. Then, the predictor, which is
not merged back as shown in Figure 1, contains pooling operations only and no parameters. This
forces the model to learn relevant auxiliary features earlier in the intermediate representations, so as

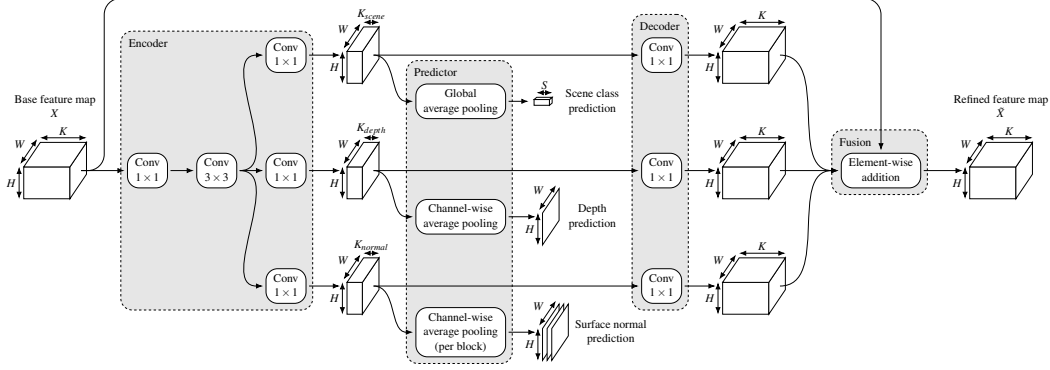

Figure 2: **Detailed architecture of residual auxiliary block (ROCK).** The block is composed of four parts, represented by the shaded areas: the encoder extracts task-specific features for all auxiliary tasks; the decoder and fusion operation transform these encodings back to the original feature space and merge them into the main path, explicitly bringing complementary information to the primary task; the predictor produces outputs for auxiliary tasks in order to learn from them through MTL. Although the block can be instantiated for any number and kind of tasks, it is presented here with the specific setup of three auxiliary tasks described in Section 3.

to maximize their influence on the primary task when they are fused into it. We evaluate our approach on object detection, with multi-modal auxiliary information (scene labels, depth and surface normals), as illustrated in Figure 1. Experiments carried out on NYUv2 dataset [44] validate the relevance of the approach: we outperform state-of-the-art detection methods on this dataset by a large margin.

## 2 ROCK: Residual auxiliary block

The general architecture of our model is shown in Figure 1. It is created from an existing model performing a given task $t_0$. This model should be composed of a backbone network yielding a base feature map $X$ (left of Figure 1), used as input to a task-specific module computing predictions (right of Figure 1). This kind of design is fairly general, so this assumption is not restrictive. The idea behind ROCK is to add a new residual auxiliary block (middle of Figure 1) between the two existing components, in order to leverage $T$ other, auxiliary tasks $\{t_i\}_{i=1}^T$ to extract useful information and inject it into the base feature map $X$ to yield a refined version $\tilde{X}$ of it. This refined representation, being similar to the base feature map, is then used by the task-specific module of the primary task, which is now explicitly influenced by auxiliary tasks. The new task-specific features might not be easily learned from the primary task $t_0$ only, so $\tilde{X}$ encodes additional details of the scenes learned by the block, therefore leading to better performance on the primary task $t_0$.

To refine the base feature map $X$, the auxiliary block must extract information from all auxiliary tasks $\{t_i\}_{i=1}^T$. To this end, it is learned within Multi-Task Learning (MTL) framework: during training, a prediction $y_t$ is produced for every task $t$ and a loss $\ell_t$ is applied, so that the block is learned from all tasks (including the main primary task, through the refinement path) simultaneously. Learned intermediate features are then used in the refinement step. In the inference phase, features are extracted for auxiliary tasks and are used to modify the base feature map in the same way, so that the predictions for the primary task explicitly take this information into account, without needing any annotations. Therefore, ROCK uses auxiliary supervision as privileged information.

We now present the general design of the residual auxiliary block for arbitrary tasks, then detail the architecture we use in the experiments on NYUv2 dataset with the associated tasks in Section 3.1. The block is thought to be generic, so that it can be easily integrated into a wide range of networks and can be applied to almost any task, without further major change. All its components are designed to have a small computational overhead, in order to keep the increase in complexity light, easing the integration of the block into existing architectures. It also has as few parameters as possible. The resulting model can therefore be learned efficiently, and fully leverage additional annotations to effectively increase performance.

Our auxiliary block is composed of four main parts: encoder, decoder, fusion and predictor. They are all illustrated in Figure 1 and detailed in Figure 2 within shaded blocks. We note that we use a simple design here to have a generic approach and show its benefits, but more complex architectures could lead to better results through better feature learning.

The base feature map $X$ is first processed by the encoder $Enc$, whose role is to learn task-specific features $\{Enc_{t_i}(X)\}_{i=1}^T$ from it with dedicated heads. For each task $t$, we use a bottleneck-like architecture to keep computation low. As shown in Figure 2, it is composed of a $1 \times 1$ convolution to reduce width of the base feature map by a factor of 4, followed by a $3 \times 3$ convolution with the same width. The last operation is a task-specific $1 \times 1$ convolution to yield a width $K_t$ adapted to the task $t$. When learning from multiple auxiliary tasks, the first two layers of the encoder are shared to have a common encoder trunk with task-specific heads, further reducing computation. The task-specific encodings obtained here are then used as input for both the decoder and the predictor, and should therefore contain all necessary information about auxiliary tasks to be used in the refinement step. We detail how this is achieved in the following.

## 2.1 Merging of primary and auxiliary representations

Different tasks bringing different kinds of supervision, the previous encodings $\{Enc_{t_i}(X)\}_{i=1}^T$ should contain information complementary to what is learned from the primary task $t_0$. Therefore, it seems useful to combine them with the base feature maps $X$ to get more complete representations of the scenes. The second and third parts of the residual auxiliary block are the decoder $Dec$ and the fusion step $F$. The former takes the output of the encoder and projects it back to the space of the base feature map, so that it can be injected back into the primary path. This is done for each task $t$ separately with a single $1 \times 1$ convolution to have the same width as the base feature map (see Figure 2). The fusion step then merges all these task-specific features with the base one in a residual manner to yield the refined feature map $\tilde{X}$, which encodes both primary and auxiliary information:

$$\tilde{X} = F\left(X, \{Dec_{t_i} \circ Enc_{t_i}(X)\}_{i=1}^T\right) = X + \sum_{i=1}^T Dec_{t_i} \circ Enc_{t_i}(X). \tag{1}$$

The residual formulation allows the base feature map to keep its content while focusing it more on relevant details of the images, yielding better features for the primary task. This feature merging step is key in ROCK to improve upon flat MTL, and these two modules are the main difference between flat and primary MTLs. In flat MTL, all tasks are at the same level and are able to benefit each other through shared feature learning only, *i.e.* their mutual influence is implicit in the models. By injecting the auxiliary representations into the primary one, we break the symmetry between tasks, effectively favoring the primary one. This task is then explicitly influenced by the auxiliary tasks through fusion, and the model can fully leverage auxiliary supervision.

## 2.2 Effective MTL from auxiliary supervision

The last element of the residual auxiliary block is the predictor $Pred$. Its purpose is to produce predictions $\{y_{t_i}\}_{i=1}^T$ for all auxiliary tasks $\{t_i\}_{i=1}^T$, so that losses can be applied to learn from the tasks through MTL. Its inputs are the feature encodings $\{Enc_{t_i}(X)\}_{i=1}^T$ from the encoder, whose sizes are already task-specific. Since the predictor might lose information with respect to these features in order to yield the predictions, only the features from the encoder go through the decoder to be merged back into the main path (as illustrated in Figure 2 and formalized in Equation (1)), so that more information is kept for use in the primary task. Therefore, all parameters learned in the predictor are to be thrown away after training, *i.e.* they are not used for inference (we are only interested in the primary task, the auxiliary tasks being used only to improve its performance). In order to force the model to learn useful information in the encoder and not in the predictor, so that it is kept and merged back, we use a predictor composed of pooling layers only, with no learned parameter:

$$y_t = Pred_t\left(Enc_t(X)\right) = Pool_t\left(Enc_t(X)\right) \tag{2}$$

with $Pool_t$ a task-specific pooling operation. The kinds of pooling used are dependent on the tasks considered, as they are directly linked to the natures of the tasks (*e.g.* scalar or spatial). Once again, this design choice of not having any learned parameter in the predictor is important for ROCK to distinguish from flat MTL. It forces the task-specific representation learning to happen within the encoder, and therefore to take part in the refinement step. This is a way to maximize the influence of the auxiliary tasks on the primary one, *i.e.* to get away from flat MTL.

# 3   Application to object detection with multi-modal auxiliary information

We incorporate ROCK for object detection as the primary task, using multi-modal auxiliary information: scene classification, depth prediction and surface normal estimation (Figure 1). The use of annotations from different but related tasks to improve performance is a common approach. Our problem is related to the use of semantic (scene) and geometric (depth and surface normals) information, which have been successfully combined [30, 27, 49, 20, 13, 8], although not directly for object detection and primary MTL. In the context of object detection, combining several additional informations, *e.g.* depth [45, 19, 18, 48, 37], or surface normal and surface curvature [48] has shown to significantly improve performance. Our problem is slightly different since we only use auxiliary information during training. More related to our context, [39] leverages depth, surface normals and instance contours to pre-train a model on synthetic data through flat MTL, then fine-tunes it for object detection on real target data. We differ from this kind of approach by our MTL strategy, which is driven by an object detection primary task.

Our approach is closely connected to [23], which uses depth as privileged information to improve object detection. However, both methods differ in the way they use the depth annotations. While [23] directly uses depth to perform object detection, we merge intermediate representations used for depth prediction. This difference leads to an earlier fusion in ROCK, where intermediate representations from all tasks are fused together to benefit from the correlation between tasks, while [23] uses a late fusion of predictions. We show in the experiments that we outperform [23] for object detection, validating the relevance of our approach.

## 3.1   Instantiation of ROCK

We now describe how ROCK is instantiated for the three tasks utilized with NYUv2 dataset [44]. This dataset contains relatively few images compared to large-scale datasets, *e.g.* ImageNet [40], so additional supervision might yield a larger gain than on bigger datasets. We also show that ROCK can handle a larger-scale synthetic dataset with missing annotation modality in Section 4.3.

For scene classification, the encoder and predictor follow the common design for classification problems: the last layer of the encoder is a classification layer into $K_{scene} = S = 27$ scene classes, and the pooling is a global average pooling, reducing spatial dimensions to a single neuron while keeping width equal to the number of classes. Error is computed with a cross-entropy loss preceded by a SoftMax layer over the $S$ classes as is common in classification tasks.

For depth estimation, annotations consist in a single depth map for each example. We choose $K_{detph}$ by dividing previous width by a factor of 4, as a trade-off between compressing maps to the final target width of 1 and keeping enough information to provide to decoder. We then use a channel-wise average pooling reducing width from $K_{depth}$ to 1, while keeping spatial dimensions. The spatial resolution of predictions is the same as that of the base feature map, and therefore depends on where the block is inserted into the network. The regression loss used here is a reverse Huber loss [28] in log space, as it has been shown to yield good results for depth prediction.

The last surface normal estimation task is similar to the depth estimation one, with the differences that ground truth maps represent normalized vectors, *i.e.* are of size 3 and $L_2$ normalized. The structure of the auxiliary block is therefore close too: we apply the same strategy as for the depth estimation task separately for each component of the vectors (*i.e.* $K_{normal} = 3K_{depth}$ and the channel-wise pooling is applied for each block of $K_{depth}$ maps) and concatenate the resulting three maps. The loss is different however: it is the sum of negative dot product and $L_2$ losses [8], following a $L_2$ normalization layer.

Once intermediate features are extracted from all auxiliary tasks, they are all fused into the base feature maps to yield its refined version. As shown in Figure 2 and in Equation (1), we do it here with a generic element-wise addition of all feature maps. The optimal fusion scheme could depend on the nature of primary and auxiliary tasks. For example, element-wise product can be interpreted as a gating mechanism [9, 22], which is well suited when the auxiliary task can be interpreted as an attention map. We show in the experiments that this fusion strategy is relevant for leveraging depth information. Finally, more complex fusion models, *e.g.* full bilinear fusion schemes [15, 2], could certainly be leveraged in our context.

# 4 Experiments

In this section, we first present an ablation study of ROCK (Section 4.1) to evaluate the effect of every component, then we compare ROCK to other state-of-the-art object detection methods on NYUv2 dataset (Section 4.2) and using another large-scale dataset (Section 4.3). We finally conduct several further experiments to finely analyze ROCK (Section 4.4).

**Experimental setup.** We use NYUv2 dataset [44] for the experiments. It is composed of an official train/test split with 795 and 654 images respectively. For model analysis and ablation study, we further divide the train set into new train and val sets of 673 and 122 images respectively, taking care that images from a same video sequence are all put into the same set. We then train our model on the train and val sets and evaluate it on the official test split for comparison with state of the art. Object detection is performed on the same 19 object classes as [19, 23] and is evaluated with three common metrics to thoroughly analyze proposed improvements. We use the SSD framework [31] with a ResNet-50 [21] backbone architecture pre-trained on ImageNet [40]. We train the networks using Adam optimizer [24] with a batch size of 8 for 30,000 iterations with a learning rate of $5 \cdot 10^{-5}$, then we lower it to $5 \cdot 10^{-6}$ and keep training for 10,000 more iterations. We use a standard setup for object detection, the details of which can be found in supplementary.

## 4.1 Ablation study

**ROCK architecture.** We present an ablation study of ROCK in Table 1 to identify the influence of each component. The first row shows results of our baseline, which is a ResNet SSD model. The last row corresponds to our full ROCK model, which yields improvements of 6.4, 1.3 and 2.3 points in all three metrics with respect to the baseline. To break down this gain between using additional supervision and using our auxiliary block, we first consider a simple flat multi-task SSD baseline, presented in the second row of Table 1. The task-specific heads applied on *conv5* feature map are just $1 \times 1$ convolution into $S$, 1 or 3 maps depending on the task, followed by a global average pooling for scene classification. This model has an improvement of 3.1, 0.2 and 1.2 with respect to the baseline, which corresponds to the use of additional annotations, *i.e.* the gain from Multi-Task Learning. Then we use our residual auxiliary block but remove the feature merging step, *i.e.* the decoder and fusion, while keeping the encoder and predictor the same. This is shown in the third row of Table 1. This results in an improvement of 1.2, 0.2 for the first two metrics with respect to the flat MTL baseline, which is specifically brought by our auxiliary block, compared to a more common way of doing MTL. The difference between our full ROCK model and this last one, *i.e.* 1.9, 0.9 and 1.1 points on all metrics, is due to the feature merging step, therefore validating the explicit exploitation of auxiliary features through fusion for object detection.

Table 1: **Ablation study of ROCK** on NYUv2 val set in average precision (%).

| | Model | | | Results | | |
|---|---|---|---|---|---|---|
| Name | Auxiliary annotations | Aux. task encoding | Feature merging | mAP@ 0.5 | mAP@ 0.75 | mAP@ [0.5:0.95] |
| Detection baseline | | | | 31.2 | 15.8 | 16.2 |
| Flat MTL baseline | ✓ | | | 34.3 | 16.0 | 17.4 |
| ROCK w/o fusion | ✓ | ✓ | | 35.7 | 16.2 | 17.4 |
| ROCK | ✓ | ✓ | ✓ | **37.6** | **17.1** | **18.5** |

**Contributions of auxiliary tasks.** In order to evaluate the importance of each auxiliary task, Table 2 presents another ablation study with results obtained when one task is dropped at a time, both for the flat MTL baseline and ROCK. It appears that all supervisions are leveraged to improve results for both models, with small differences in their contributions.

## 4.2 Comparison with state of the art

We compare ROCK to other state-of-the-art object detection methods on NYUv2 dataset in Table 3. The first two entries ([19, 48]) of the table use detection annotations only. It is noticeable that all

Table 2: **Ablation study of auxiliary supervisions** for flat MTL baseline (left) and ROCK (right) on NYUv2 val set in average precision (%). Auxiliary supervision used is given between parentheses (D: depth, N: surface normals, S: scene class).

| Name | mAP@ 0.5 | mAP@ 0.75 | mAP@ [0.5:0.95] | Name | mAP@ 0.5 | mAP@ 0.75 | mAP@ [0.5:0.95] |
|---|---|---|---|---|---|---|---|
| Flat MTL (DN) | 32.1 | 15.9 | 16.1 | ROCK (DN) | 34.0 | 16.6 | 16.8 |
| Flat MTL (DS) | 32.7 | 15.9 | 16.3 | ROCK (DS) | 33.2 | 16.1 | 16.3 |
| Flat MTL (NS) | 32.7 | 15.8 | 16.1 | ROCK (NS) | 35.1 | 16.8 | 17.1 |

other methods, leveraging some kind of additional information, outperform them by a large margin, indicating that augmenting images with more annotations has a large impact on this dataset with few examples. Our ROCK model outperforms Modality Hallucination network [23] by 3.1 points in the same setting, where only depth is used as privileged information. This validates that our approach is able to exploit correlations between depth estimation and object detection. ROCK is also competitive with methods using depth during inference too (*i.e.* not as privileged information) ([48]), even when they are trained on additional synthetic data ([19]), as displayed on the following two rows.

Using more annotations yields significantly better results again, as shown with the use of surface normal and curvature ([18, 48]). When ROCK adds supervision from surface normal estimation and scene classification, results are greatly improved, by 2.7 points with respect to using depth only. By specifically designing the architecture to leverage this auxiliary supervision to improve the primary object detection performance, ROCK even outperforms methods using similar kinds of annotations, but at test-time too, in contrast with the privileged context of ROCK.

Table 3: **Detailed detection results on NYUv2 test set** in average precision (%) with an IoU threshold of 0.5. Additional supervision used for training is indicated between parentheses (D: depth, N: surface normals, C: surface curvature, S: scene class). A $^\star$ means that additional information is also used during inference. Methods marked with (+SYN) and (+MLT) are trained with additional synthetic data (see [19] or [18] for details) and pre-trained on MLT dataset [51] respectively.

| Model | mAP | btub | bed | bshelf | box | chair | counter | desk | door | dresser | gbin | lamp | monitor | nstand | pillow | sink | sofa | table | tv | toilet |
|---|---|---|---|---|---|---|---|---|---|---|---|---|---|---|---|---|---|---|---|---|
| RGB R-CNN [19] | 22.5 | 16.9 | 45.3 | 28.5 | 0.7 | 25.9 | 30.4 | 9.7 | 16.3 | 18.9 | 15.7 | 27.9 | 32.5 | 17.0 | 11.1 | 16.6 | 29.4 | 12.7 | 27.4 | 44.1 |
| RGB R-CNN [48] | 22.8 | 16.2 | 41.0 | 28.0 | 0.7 | 27.4 | 34.6 | 8.4 | 15.2 | 16.9 | 16.5 | 25.9 | 38.4 | 12.1 | 15.0 | 27.5 | 28.2 | 10.6 | 24.9 | 44.8 |
| Modality Hallucination (D) [23] | 34.0 | 16.8 | 62.3 | 41.8 | 2.1 | 37.3 | 43.4 | 15.4 | 24.4 | 39.1 | 22.4 | 30.3 | 46.6 | 30.9 | 27.0 | 42.9 | 46.2 | 22.2 | 34.1 | 60.4 |
| ROCK (D) [ours] | 37.1 | 23.5 | 61.8 | **43.0** | 1.5 | 51.8 | 42.5 | 19.5 | **35.7** | 22.9 | 39.0 | 39.8 | 40.0 | 37.7 | 38.5 | 36.6 | 49.8 | 22.0 | 47.1 | 53.1 |
| RGB-D R-CNN (D$^\star$) [48] | 35.5 | 37.8 | 69.9 | 33.9 | 1.5 | 43.2 | 45.0 | 15.7 | 20.5 | 32.9 | 32.9 | 33.7 | 50.9 | 31.6 | 37.3 | 39.0 | 49.0 | 22.9 | 32.2 | 44.9 |
| RGB-D R-CNN (D$^\star$+SYN) [19] | 37.3 | 44.4 | 71.0 | 32.9 | 1.4 | 43.3 | 44.0 | 15.1 | 24.5 | 30.4 | 39.4 | 36.5 | 52.6 | 40.0 | 34.8 | 36.1 | 53.9 | 24.4 | 37.5 | 46.8 |
| Pose CNN (DN$^\star$+SYN) [18] | 38.8 | 36.4 | 70.8 | 35.1 | **3.6** | 47.3 | 46.8 | 14.9 | 23.3 | 38.6 | **43.9** | 37.6 | 52.7 | 40.7 | 42.4 | 43.5 | 51.6 | 22.0 | 38.0 | 47.7 |
| RGB-Geo R-CNN (DNC$^\star$) [48] | 39.3 | 41.8 | 75.0 | 36.4 | 2.2 | 46.9 | 46.4 | 15.8 | 23.9 | 37.9 | 39.9 | 37.5 | 53.0 | 41.7 | 44.0 | 44.4 | 51.8 | 26.9 | 34.5 | 47.0 |
| ROCK (DNS) [ours] | 39.8 | 22.7 | 66.9 | 40.0 | 3.2 | 51.5 | 41.8 | 16.6 | 33.7 | 34.7 | 37.4 | **43.3** | 38.8 | 47.0 | 41.7 | 43.8 | 52.1 | 23.7 | 53.7 | 63.3 |
| ROCK (DNS+MLT) [ours] | **46.8** | **45.8** | **77.4** | 40.8 | 3.2 | **60.2** | 48.4 | **30.1** | **35.7** | 42.6 | 43.1 | 39.7 | **54.3** | **60.4** | 45.4 | 44.9 | **63.0** | 32.5 | **55.0** | **66.2** |

## 4.3 Pre-training on large-scale MLT dataset and Fine-Tuning

To test ROCK in a more challenging context, we pre-train it on MLT dataset [51], then fine-tune it and evaluate it on NYUv2 dataset. MLT is composed of over 500,000 synthetic indoor images similar to these from NYUv2, and annotated for object detection, depth and surface normal estimation. This makes this dataset well suited for Fine-Tuning (FT) to NYUv2. However, it raises three main challenges. First, the scale of the dataset is several orders of magnitude larger. In contrast to NYUv2, MLT images are synthetic, so FT requires to address the domain shift between the two datasets. MLT also does not provide scene classes and ROCK would then have to handle imbalance between pre-trained and newly added tasks when transfered from MLT to NYUv2. We keep the same setup as before but ROCK is learned on 23 slightly different object classes, for 240,000 and 80,000 iterations with the same learning rates. The scene classification branch is removed as there is no annotation for it. Results are presented in the last row of Table 3. FT from MLT to NYUv2 gives an outstanding state-of-the-art performance of 46.8 points, which is an improvement of 7.0 points over directly training on NYUv2. This result shows that ROCK is able to overcome challenges associated with MLT dataset, in particular to scale to larger datasets and to handle heterogeneous data and missing annotation modalities.

### 4.4 Further analysis

**Complexity of ROCK.**    We conduct an analysis of the complexity of ROCK with ResNet-50 as backbone architecture. A comparison of numbers of parameters and inference times without and with ROCK is displayed in Table 4. It shows that including ROCK into the network only yields a slight increase in complexity (around 17% more parameters and 7% slower in time), easing its integration into existing models.

Table 4: **Complexity of ROCK** in parameters and inference time measured with ResNet-50 backbone.

| Model | Number of parameters | Inference time (ms/image) |
|---|---|---|
| Detection baseline | 27.8M | 57 |
| ROCK | 32.6M | 61 |

**Analysis of architecture of residual auxiliary block.**    We present several design experiments to validate the architecture of our residual auxiliary block in Table 5. We first verify that the performance improvement of ROCK is not due to the additional parameters introduced in the model. For this, we evaluate ROCK with the complete architecture but with all auxiliary loss weights set to 0, effectively deactivating Multi-Task Learning. The results are shown on the left of Table 5 and are close to those of the detection baseline, indicating that the auxiliary block is only useful to learn from auxiliary tasks in an effective way. We study the effect of the fusion operation with depth only on the right part of Table 5. It appears that the product is superior to the addition for this task. Depth bringing a geometric information, the product can be interpreted as a spatial selection. However, design of this component has not been fully explored and further experiments should yield better results.

Table 5: **Analysis of architecture of ROCK** on NYUv2 val set in average precision (%).

| Model | mAP@ 0.5 | mAP@ 0.75 | mAP@ [0.5:0.95] | Model (depth-only) | mAP@ 0.5 | mAP@ 0.75 | mAP@ [0.5:0.95] |
|---|---|---|---|---|---|---|---|
| ROCK | **37.6** | **17.1** | **18.5** | El.-wise addition | 30.9 | 14.8 | 16.1 |
| ROCK w/o aux. sup. | 30.6 | 15.6 | 16.2 | El.-wise product | **32.3** | **16.2** | **17.3** |

**Effectiveness of additional supervision.**    We here analyze the relation between getting more images or additional annotations on available images. To this end, we train ROCK on a fraction of the train set and observe how many examples are needed to get the same performance as the detection baseline (*i.e.* without auxiliary supervision) on the whole train set. Results are summarized in Table 6. Training ROCK on around 70% of the train set roughly gives similar results than the detection baseline (depending on which metric is used for comparison), *i.e.* having the additional three auxiliary tasks to learn from compensates for the loss of 30% of examples. This result shows that fully annotating available data with more tasks can be helpful in domains where examples are hard to obtain.

Table 6: **Effectiveness of additional supervision** on NYUv2 val set in average precision (%).

| Model | mAP@ 0.5 | mAP@ 0.75 | mAP@ [0.5:0.95] |
|---|---|---|---|
| Detection baseline (on 100% of train set) | 31.2 | 15.8 | 16.2 |
| ROCK on 60% of train set | 29.5 | 12.2 | 13.9 |
| ROCK on 70% of train set | 32.8 | 14.5 | 15.9 |
| ROCK on 80% of train set | 34.7 | 16.2 | 17.0 |

**Visualization of results.**    We show outputs of ROCK on some unseen images in Figure 3 for qualitative visual inspection. In the first row, the baseline model wrongly detects a table. However, the classification of the scene into the *bathroom* class might decrease the probability of such an object class, in favor to classes seen more often in these scenes. It is noticeable that detections produced by ROCK agree more with the scene class. On the second row, ROCK detects more objects than the baseline, especially the bed which is only partially visible. This may be due to the depth prediction,

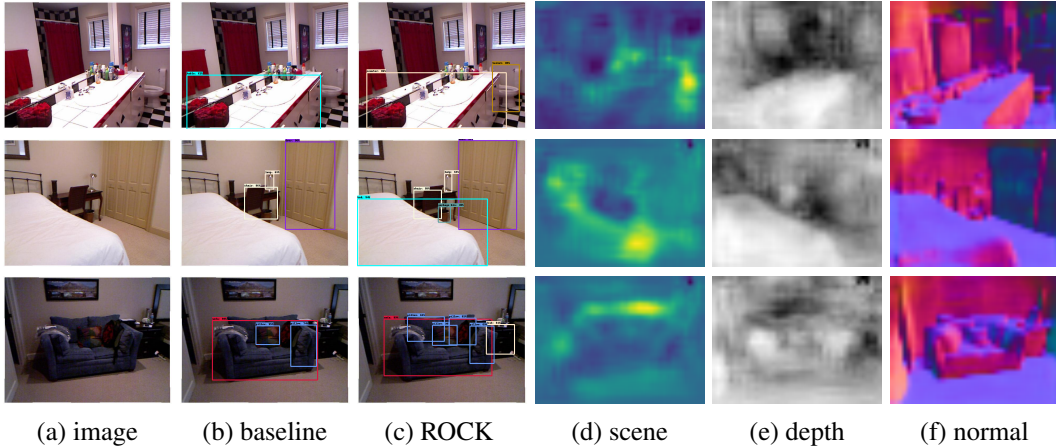

| (a) image | (b) baseline | (c) ROCK | (d) scene | (e) depth | (f) normal |

Figure 3: **Visualization of outputs.** The original images are presented in (a). Outputs of the detection baseline and ROCK are illustrated in (b) and (c) respectively. Column (d) depicts scene classification through heatmaps of ground truth scene classes (*i.e.* the maps just before global average pooling). Columns (e) and (f) show predictions for depth prediction and surface normal estimation respectively.

where a clear separation of the bed from the rest of the scene is present, easing its detection. In the last row, the pillows are rather difficult to distinguish as they all have similar colors. The surface normal prediction brings geometric information enabling to discern instances and find their contours more easily, leading to better detections. Additional examples are presented in the supplementary.

**Generalization to another dataset.**    To evaluate the generality of the approach, we run additional experiments on CityScapes dataset [6], which is composed of outdoor scenes in urban context, in order to contrast with NYUv2. We train ROCK on it for object detection (8 object classes), with disparity estimation as auxiliary task, using a similar setup (with the same configuration as for depth estimation, but with 60,000 training iterations). We use the train set for learning and evaluate on the val set. Results are shown in Table 7. Again, ROCK outperforms the detection baseline by 1.2, 0.9 and 1.1 points in all metrics, showing the generality of our approach.

Table 7: **Results of ROCK on CityScapes dataset** [6] val set in average precision (%).

| Model | mAP@ 0.5 | mAP@ 0.75 | mAP@ [0.5:0.95] |
|---|---|---|---|
| Detection baseline | 42.8 | 19.0 | 21.6 |
| ROCK | **44.0** | **19.9** | **22.7** |

# 5    Conclusion

In this paper, we introduced ROCK, a generic multi-modal fusion block for deep networks, to tackle the primary MTL context, where auxiliary tasks are leveraged during training to improve performance on a primary task. By designing it with a residual connection and intensive pooling operators in predictors, we maximize the impact and complementarity of the auxiliary representations, benefiting the primary task. We apply ROCK to object detection on NYUv2 dataset and outperform state-of-the-art flat MLT by a large margin. We show that exploiting additional supervision with ROCK yields the same performance than having around 30% additional examples with a single-task model, encouraging to fully exploit available data in contexts where images are difficult to gather. By pre-training our model on a large-scale synthetic dataset with different classes and auxiliary modalities, we set a new state of the art on NYUv2 and demonstrate ROCK is flexible and can adapt to various challenging setups. However, the design of ROCK has been kept fairly simple to prove the relevance of the approach. In particular, the fusion operation could be studied more thoroughly.

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
