[Supplementary Material · 687_supplementary.pdf]

# Revisiting Multi-Task Learning with ROCK: a Deep Residual Auxiliary Block for Visual Detection Supplementary Material

**Taylor Mordan**[(1, 2)]
taylor.mordan@lip6.fr

**Nicolas Thome**[(3)]
nicolas.thome@cnam.fr

**Gilles Henaff**[(2)]
gilles.henaff@fr.thalesgroup.com

**Matthieu Cord**[(1)]
matthieu.cord@lip6.fr

(1) Sorbonne Université, CNRS, Laboratoire d'Informatique de Paris 6, LIP6
F-75005 Paris, France

(2) Thales Land and Air Systems
2 Avenue Gay-Lussac, 78990 Élancourt, France

(3) CEDRIC, Conservatoire National des Arts et Métiers
292 Rue St Martin, 75003 Paris, France

## 1  Model architecture details

Our model is based on a ResNet-50 backbone with a dilation in the last block. ROCK block is applied on the *conv5* feature map, whose spatial dimensions are divided by 16 with respect to the input image (*i.e.* $30 \times 40$ for $480 \times 640$ inputs) with a width of 2048. The encoder is composed of two convolutions that are shared across all auxiliary tasks ($1 \times 1$ and $3 \times 3$ convolutions, both into 512 channels). The last convolution is $1 \times 1$ but the number of output channels is task-dependent. It is $S = 27$ for scene classification, 128 for depth estimation, and $3 \times 128$ for surface normal estimation. The predictor is composed of pooling only. It is a global average pooling for scene predictions, and a channel-wise average pooling (*i.e.* reducing the width to 1) for depth predictions and for each one of the three components of surface normal predictions. The decoder consists in a $1 \times 1$ convolution into 2048 channels for each task separately. All convolutions are followed by Batch Normalization [5] and ReLU modules, except for those just before predictors (last task-specific convolution of the encoder) that have linear activation functions. We then use a SSD detection block [9] with 6 additional prediction feature maps, learned with the same setup as in the original paper [9].

## 2  Experimental setup

Object detection is performed on the same 19 object classes as [2, 4] and is evaluated with three common metrics (mAP@0.5, mAP@0.75 and mAP@[0.5:0.95]) to thoroughly analyze proposed improvements. As additional auxiliary tasks, we use scene classification into $S = 27$ scene classes, depth estimation and surface normal estimation. The ground truths for the first two tasks are provided along with NYUv2 dataset, and we use the targets computed by [11] for the normal prediction task.

We use the SSD framework [9] with a ResNet-50 [3] backbone architecture pre-trained on ImageNet [10]. Detection is performed on the output of the *conv5* block of ResNet (or its refined version when using our auxiliary block) and on 6 additional feature maps randomly initialized. We train the networks using Adam optimizer [6] with a batch size of 8 for 30,000 iterations with a learning

| (a) image | (b) baseline | (c) ROCK | (d) scene | (e) depth | (f) normal |

Figure 1: **Visualization of outputs.** The original images are presented in (a). Outputs of the detection baseline and ROCK are illustrated in (b) and (c) respectively. Column (d) depicts scene classification through heatmaps of ground truth scene classes (*i.e.* the maps just before global average pooling). Columns (e) and (f) show predictions for depth prediction and surface normal estimation respectively.

rate of $5 \cdot 10^{-5}$, then we lower it to $5 \cdot 10^{-6}$ and keep training for 10,000 more iterations. We use the data augmentation from SSD [9] but with fixed aspect ratio for the crops. All examples are then resized to $480 \times 640$ pixels, flipped with probability 0.5, and some color data augmentation [7] is finally applied. Annotations for depth and surface normal estimation are modified accordingly to keep geometries of the scenes (see [1] for details). Classification and localization losses have weights of 1 and 3 respectively. Loss weights of auxiliary tasks are set to 3 for scene classification and depth estimation, and to 30 for normal estimation (we use a factor of 10 for this task as advised by [1]). Finally, we use the same matching strategy and classification prior as [8], and post-process detections with NMS using a threshold of 0.3.

## 3 Visualization of results

Additional visualizations are given in Figure 1.