[Reviews · NeurIPS 2018]

Reviewer 1



Summary: The paper presents a simple yet effective learning framework for object detection with auxiliary task learning. The proposed framework, ROCK, introduces a residual block that refines the base feature map using augmented representations trained with respect to corresponding auxiliary tasks. Individual decoded feature maps for each task are then fused with the base feature map by element-wise addition and the refined feature map is used for object detection. The paper provides ablation study by breaking down each component from the proposed ROCK method and shows effectiveness of the components. Overall, the proposed framework demonstrates substantial improvement on object detection on NYUv2 dataset. Strength: - The paper is clearly written and seems complete. - The proposed method is simple yet authors demonstrates its effectiveness. Comment: - It would be good to provide some ablation study how to tune hyper-parameters that balances between different loss (e.g., detection loss, scene class prediction loss, depth-prediction loss, and surface-normal prediction loss). - As the paper presents a connection to LUPI framework, it would be good to provide some results when auxiliary supervision is available at "test" time by replacing encoder of ROCK auxiliary supervision input. This may give a performance upper bound to the proposed framework. - How generalizable the proposed framework and the choice of auxiliary loss to different tasks?

Reviewer 2



Main Idea: The work targets a special case of multi-task learning (MTL) referred to as primary MTL where the goal is to use auxiliary tasks to improve performance on a primary task. The primary task considered is object detection with scene classification, depth prediction, surface normal estimation being the auxiliary tasks. The authors introduce a Residual Auxiliary Block (ROCK) which can be used to augment existing deep network-based object detection pipelines. ROCK is proposed as an alternative to the standard MTL approach used in object detection literature where a backbone feature extractor is followed by multiple task-specific heads e.g. classification, bounding box regression, and segmentation heads in MaskRCNN. The paper comprises of justification of various design choices in ROCK and ablation studies to quantify the benefit of those choices. Strengths: - The authors do a good job of zooming in on a specific problem (primary MTL) that they are trying to solve. Writing and experiments are appropriately designed around that problem. - Fig 2 summarizes the model well. - The writing is clear and Sec. 2 and 3 helps the reader understand the authors' motivation behind various design choices that went into ROCK. - Ablations in Tables 1 and 3 demonstrate the benefits of using auxiliary annotations, encoders, and feature fusion in ROCK over the standard MTL baseline on the NYUv2 dataset. - Table 4 tries to answer an interesting question - Is it better to have more images with one kind of annotations or fewer images with annotations for a wide variety of tasks? The numbers indicate that 70% of the data with multiple task annotations is sufficient to achieve performance comparable to a detector trained on 100% data without MTL. Weaknesses: Writing - Primary MTL should be differentiated from alternative goals of MTL (such as improving the performance of all tasks or saving memory and computational cost by sharing computation in a single network) early on in the abstract and introduction - In the abstract, a point is made about residual connections in ROCK allowing auxiliary features to explicitly impact detection prediction. This needs to be contrasted with standard MTL where the impact is implicit. - The claim about ROCK handling missing annotations modalities is unclear. The MLT dataset needs to be described in Sec 4.3. - The introduction describes Transfer Learning (TL) and Fine-tuning (FT) as sequential MTL. I do not completely agree with this characterization. TL is a broader term for the phenomenon of learnings from one task benefitting another task. Fine-tuning is a sequential way of doing it. Standard MTL is a parallel means to the same end. - The first 2 paragraphs of introduction read like literature review instead of directly motivating ROCK and the problem that it solves. - Need to describe Flat MTL and include a diagram similar to Fig 2 that can be directly visually compared to ROCK. - Fig 1 is not consistent with Fig 2. Fig 2 shows one encoder-decoder per auxiliary task whereas Fig 1 shows a single shared encoder-decoder for multiple tasks. - L88-89 try to make the case that ROCK has similar complexity as the original model. I would be very surprised if this true because ROCK adds 8 conv layers, 3 pooling layers, and 1 fusion layer. Inference timings must be provided to make this claim. Weaknesses: Experiments - Ablation showing the contribution of each auxiliary task on object detection performance in ROCK as well as in standard MTL. This can be done by comparing the full model (DNS) with models where one task is dropped out (DS, NS, DN). - Cite MLT dataset in Table 2 caption and describe Geo, presumably some kind of geometric features. - More details are needed for reproducibility like backbone architecture, activations between conv layers, number of channels, etc. Weaknesses: Originality - While primary MTL setting considered in this work is very useful, especially in data-deficient domains, the main contribution of this work, the ROCK architecture, comes across as a little incremental. Summary My current assessment is that in spite of the weaknesses, the model is simple, reasonably well motivated and shows decent performance gains over appropriate baselines. Hence I am currently leaning towards an accept with a rating of "6". I am not providing a higher rating as of now because the writing can be improved in several places, an ablation is missing, and the novelty is limited. My confidence score if 4 because I have only briefly looked at [19] and [35] which seem to be the most relevant literature. --Final Review After Rebuttal-- The authors did present an ablation showing the effect of different tasks on multitask training as requested. A significant number of my comments were writing suggestions which I believe would improve the quality of the paper further. The authors have agreed to make appropriate modifications. In spite of some concern about novelty, I think this work is a valuable contribution towards an understanding of multitask learning. All reviewers seem to be more or less in agreement. Hence, I vote for an accept and have increased my rating to 7 (previously 6).

Reviewer 3



This paper presents a Multi-Task based Residual Auxiliary block that modifies the output feature map of classical CNN networks (Resnet, ...) to improve it for an object detection application. The auxiliary tasks used here are optical flow and depth but the method is generic and other kinds of task may be used. This auxiliary block is a kind of feature-map auto-encorder driven by the auxiliary tasks. The main idea is that using auxiliary tasks, we can improve the generalisation of the main detection task. The paper is easy to read and well-written. The residual Auxiliary block, that is the main contribution of the paper, is composed by light elements in order to reduce the cost and weight of the network. However, the authors don't give any information on this cost in the experimental part but they only give information on the mAP. It should be interesting: 1) to give this information and 2) to try more complex structures into the residual block in order to show if it possible to improve the mAP. Experiments show that the proposed MTN can improve MAp significantly on the NYUv2 detection dataset. It should be interesting to give information about the evolution of the loss during the training. Is the final loss of the detection lower when using ROCK? Comparinf the loss of the training and validation sets should also helps to understand id the residual block really helps to generalize.